# Use of 3D Inferred Imagining for Detection of Changes in Geology in Longwall-Type Excavation Front

Paulina Lewińska [1,2]

1 Faculty of Geo-Data Science, Geodesy, and Environmental Engineering, The AGH University of Krakow, Aleja Mickiewicza 30, 30-059 Kraków, Poland; lewinska.paulina@gmail.com
2 Department of Computer Science, University of York, Heslington, York YO10 5DD, UK

**Abstract:** In this paper, I will show and describe a method of integrating infrared images with a 3D model of the front of an excavation in the longwall type of workings. I will also test the created 3D model for its potential usefulness in geological prospecting, used for looking for changes in geological layout at the front of the excavation. Geological information on the amount of coal in the front is important for the economic side of the excavation. The mine does not want to take out waste rock, but mostly for the safety of operations. The longwall shearers and plows are not designed for excavation in extremely changeable conditions, so if too much shale appears, this might change the speed and economic results of excavation. In addition, if a fold appears, this can destroy the excavating complex. Currently, the geological survey requires a geologist to get to the front, often to the unsupported roof part of the wall, clean the front and sample, and measure the geology. This is dangerous for the geologist. Thus, remote, infrared measurements would improve the safety of the staff and allow the survey to be taken in more places along the longwall. In this paper, I will also propose how such a system could be implemented and what the limitations are.

**Keywords:** remote sensing; IR; mining; point cloud





## 1. Introduction

The use of infrared images (IR) has been a recurring topic in the underground mining industry [1–11]. There are several common uses of infrared technology in above-ground mining operations, such as monitoring coal spoil tip fires and inspecting conveyor belts and electrical elements. The monitoring of both coal spoil tips and heaps is particularly prominent, as it can be performed with on-ground IR cameras, UAVs (unmanned aerial vehicles) with IR cameras, and also satellite IR images [12–16]. This is also an important topic since it allows for the detection of fires significantly faster than any other method. In the case of spoil tips, this can be performed even a few years earlier, as spoil tips burn slowly due to an oxygen-deprived environment and do not show signs unless the fire is close to the surface [17–19]. Early detection allows for putting out fires with measures such as cutting off oxygen to the areas, preventing the stack effect, or isolating the object from the source of water. The inventory of electrical parts, such as cables and batteries, is also performed with the use of IR cameras. In this case, it affects the safety of the staff more since, as a remote sensing method, there is no need for inspectors to touch or be in close proximity to the object to assess its state [11,20].

The use of infrared cameras and sensors has been used with limited success for underground excavation workings. This is due to many concerns. IR camera is an electric device that needs to undergo the same safety procedures as other equipment. Thus, it should be safe to use in potential methane- and coal dust-filled environments [21–23]. This implies the use of particular materials during the construction and dustproofness proofing. Currently, proper certificates can be obtained, and adjustments to construction can be made, but it has been a problem in the early days of IR and is still a part of safety or operation

procedures. The next problem is the ever-changing environment of the mine. This will be explained in mathematical detail in further sections of this paper, but it is important to know that IR cameras do not measure the temperature of the object; they record the temperature from the sensor to the object. This means that if the air between the camera and the object has a significant shift in temperature (such as a draft of fresh air), this will obscure the reading. The camera needs to be properly calibrated for such an environment, which may not always be possible [24,25]. All of those limitations, aided by safety regulations in the mining industry around the world, have made the use of IR cameras less common than they have the potential to be.

The current use of IR cameras and sensors varies between types of mining, methods of excavation, and preparation workings. They are often used, with varying degrees of success, to evaluate the state of electric elements in the mine [26]. However, the quality of the results of such investigations is strongly bound to the quality of the camera purchased (pixel size, the maximum/minimum temperature that the receptors can record, calibration procedure, palettes, etc.). Another prominent use is during the excavation of new workings, where IR markers can be used to show the direction of the excavation [27]. Moreover, IR can be used to detect cracks in conveyor belt tapes that would allow changing the tape before it gets damaged to a point where it snaps, creating a danger to humans and infrastructure [5,8,26]. Currently, some work is being performed on using Computer Vision and Machine Learning for the detection of human movement in the mine. This is an interesting first step in providing autonomy of operations of at least some of the machines used in the underground mine since it provides safety for the mining personnel [1,2]. However, the most important use from the point of view of human and infrastructure safety is monitoring and early detection of potential underground fires in coal or sulfide mines [28,29]. The coal mine fires are a life-threatening danger to the miners. Since their origin varies from endemic to accidental, they are hard to predict and even harder to put down. However, currently, research is in the experimental phase.

In this study, the use of infrared cameras to monitor changes in the geology at the front of a longwall-type coal excavation is proposed [4]. This is an important topic due to the weight that this information carries and the risks that one must undertake to obtain it. In longwall coal mining, mechanical–hydraulic supports support the roof just above the conveyor upon which the longwall shearers or plows are placed. This means that the area just next to the front is often not supported. This is a safety issue since the roof might collapse due to the layered nature of sedimentary rock deposits. This is usually not a problem since miners do not work in this area. However, the geological crew needs to inventory the front [23,30,31]. This is usually performed by walking up to the front of the excavation with the unsupported roof, cleaning the front from the coal dust, and making measurements with a tape measure. This process is carried out at various intervals along a longwall a few times a week. As described, this procedure is dangerous to the staff, and it also provides low-quality results from the geometry point of view since the tape measure is just set to have 0 on the rubble floor. This task is necessary for the operations of the mine to monitor the changing geology as the longwall type of excavation proceeds. This is not only important for the economic side of the excavation; the mine does not want to take out waste rock, but mostly for the safety of operations. The longwall shearers and plows are not designed for excavation of extremely changeable conditions, so if too much shale appears, this might change the speed of excavation [20]. Moreover, if a fold appears, this can destroy the excavating complex and create a danger to the workers. Thus, remote infrared measurements would improve the safety of the staff and allow the survey to be taken in more places along the long wall. In addition, connecting the IR images with any type, even a generalized 3D model of the current shape of excavations, would provide a more accurate representation of the geometry of the geology at the excavation front.

The research carried out by the author had two goals: to determine whether it is possible to use a thermal imaging camera in the conditions of the mine environment in the

longwall excavation area and whether it is possible to distinguish shale interbedding from a coal deposit on the thermographs obtained in this way.

## 2. Methods and Materials

### 2.1. Mathematical and Physics Background of Infrared Thermography

Thermal radiation is an effect of the vibration of molecular covalent bonds of matter creating an object. It is the reason why bodies at different temperatures are capable of heat exchange [32,33], described by laws of thermodynamics. Every object radiates an amount of energy produced by the vibration of the molecular covalent bonds of the combination of substances from which it is composed. Emitted energy exists in the form of discrete quanta, or photons, of energy $E_{ph}$ equal to the protons' frequency $f_{ph}$ multiplied by Planck's constant (h = 6.626 × 10 − 34 J s). This is inversely proportional to the wavelength $\gamma$, with a velocity c in a given medium. This is represented by

$$E_{ph} = hf_{ph} = h\frac{c}{\gamma} \tag{1}$$

This general equation needs to be properly used since its interpretation is crucial for all energy measurements. Both c and $\gamma$ are dependent on the type and size of medium through which the wave travels. This is why the entire electromagnetic spectrum has been divided arbitrarily into intervals, or spectral bands, from $\gamma \in (0,\infty)$ in order to calcify the received energy signatures [34,35].

The amount of emitted radiation depends on the type of the emitting surface and its temperature. Detectors in infrared cameras detect electromagnetic energy $E_t$. However, the detector has no means of checking the source of this energy: thus, it detects energy emitted by object $E_0$ and also the reflected energy coming from the surroundings $E_r$ and $E_a$, the energy emitted by the atmosphere (Figure 1). This makes the equation more complex:

$$E_t = (\varepsilon_0\tau_a)E_0 + (1 - \varepsilon)\tau_a E_\gamma + (1 - \tau_a)E_a \tag{2}$$

where:

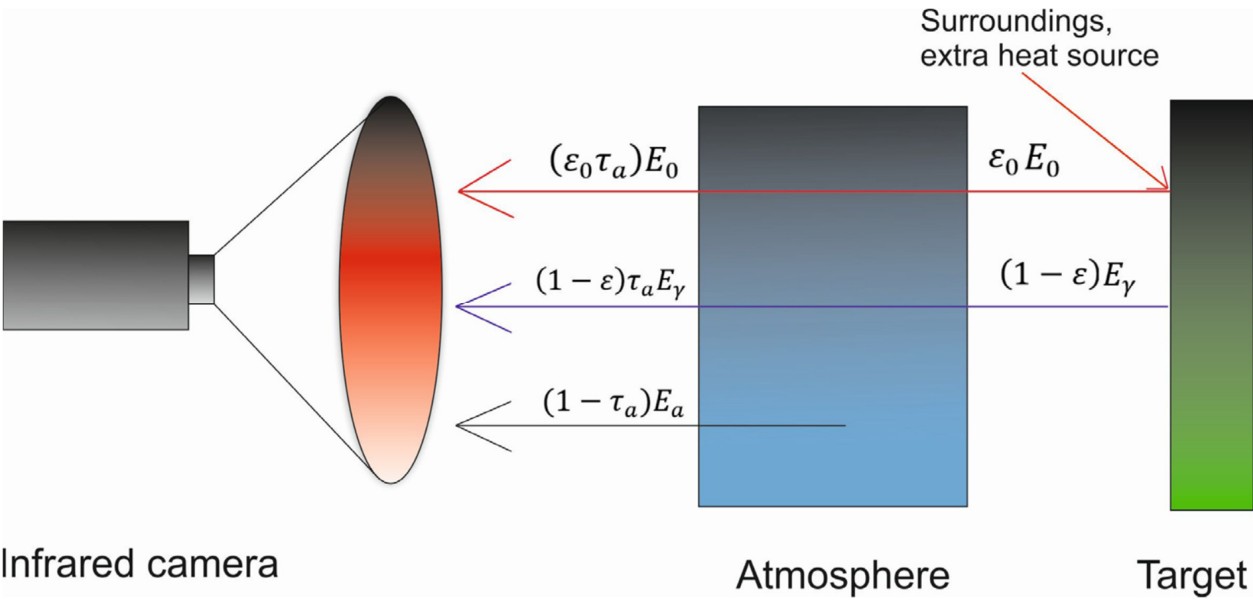

**Figure 1.** The visualization of the energy read by the receptors inside inferred camera [24].

$\varepsilon$—emissivity, is the ability of the object to radiate infrared radiation calculated using a ratio of surface radiative ability at a set temperature and the radiative ability of a blackbody under the same environmental conditions,

$\tau$—transmittance, the ratio of the transmitted light to the incident light, in respect of infrared radiation.

Depending on the object's surface characteristics and the composition below the surface, the energy also in the form of the heat reaching an object can be absorbed, reflected, or transmitted. This means that the value to be calculated is usually performed accurately only in a controlled setting, where no additional energy sources exist [3,24]. A radiation analysis is conducted in three areas of the infrared spectrum NIR (near infrared) 0.78–3 μm; MIR (mid-infrared) 3–50 μm; and FIR (far-infrared) 50–1000 μm. However, it is further divided into the following areas: near-infrared (0.7–1.4 μm), short-wave IR (1.4–3 μm), medium-wave IR (3–8 μm), long-wave IR (8–15 μm), and far infrared (15–1000 μm). Almost all civilian cameras work in the long-wave IR spectrum. Long-wave IR spectrum cameras absorb part of the atmospheric absorption, allowing for easier work.

In order to use Equations (1) and (2), it is necessary to use the emission of a black-body radiator [32,36]. However, most physical objects are described as 'grey bodies' since they incorporate elements of different types of radiation. This forces the use of monochromatic emissivity, which is a ratio of the monochromatic emissive power of the body to the monochromatic emissive power of a blackbody at the same wavelength and temperature [37,38]. By defining those conditions, an equation for the 'physically observed emissivity' $\varepsilon_{poe}$ [39], can be given by

$$\varepsilon_{poe}(\theta, \varphi) = \sum_{k=1}^{N} \alpha_k \varepsilon_k(\theta, \varphi) + K_\lambda(T_0) \sum_{k=1}^{N} \alpha_k \varepsilon_k(\theta, \varphi) \Delta T_k, \tag{3}$$

where:

$N$—a flat pixel composed of $k$ homogenous parts;

$T_0$—reference temperature that is independent of viewing direction and wavelength;

$\Delta T_k$—temperature difference between the temperature of each $k$ and the reference temperature $T_0$;

$\theta, \varphi$—vertical and horizontal viewing angles, respectively;

$\alpha_k$—relative area of each $k$ where the sum of all $\alpha_k$ are equal to 1,

$\varepsilon_k(\theta, \varphi)$ is the emissivity for each $k$ [32,40].

Equation (2) introduces the energy emitted by the atmosphere. This accounts for not only the raw, basic atmosphere but also its shifts over time (daily sun movement). This means images taken on the same day but at different times will be influenced differently by the atmosphere, making the images "look" different and giving different 'reading' of the temperature. This needs to be taken into account during the interpretation of data and during planning if the passive method is being used. Equation (3) emphasizes that measured objects are composed of many different emissivity materials. Thus, defining the value of monochromatic emissivity in field conditions is difficult, so the post-processing procedure is required. In addition, Equation (3) introduces both vertical and horizontal viewing angles, so distortions in both directions are different [24,25,41,42]. Experiments also suggest that not only the general environment needs to be taken into account but also the distance to the object and the placement of the IR camera relative to the object [43]. Images taken from the side are influenced differently by the object and environmental emissivity since the energy needs to travel a different distance before it reaches the camera receptor [30,44,45].

## 2.2. Challenges of Use of Infrared Cameras in Underground Mine Environment

In mining conditions, all of the aforementioned factors need to be taken into account, but the unique underground conditions make the survey even more difficult. The first difficulty is the environment itself. During surveys outside the mine, it is possible to choose a day when there is no rain or mist obscuring the view of the object and introducing emissivity from the water in the air. However, in the mine, the environment is more stable. That means that if the mine or part of the mine is 'wet', it is going to be 'wet' always. In addition, most of the mines are dusty. This can be coal dust (of very low emissivity), shale

dust, or salt dust (of changeable emissivity). The amount of dust might change depending on the pace of the survey, but also if the machines are operational and if the haulage is being performed. This additional source of temperature needs to be taken into account during post-processing.

Additionally, in order for mines to operate, they must be ventilated. Air is distributed from the main shaft or shafts and moved around the active workings. Then, whoever is left is directed to the exhaust shaft. The mine is also separated into sections by air dams (a type of large, heavy door) that allow for limiting or straightening the air in the section. This means that different parts of the mine have different air movements and, thus, different environments. Additionally, if a dam is shut, this might instantly change the airflow; the same happens when a large vehicle moves. The excavation fronts, depending on the method used, might be covered with water used for cooling the machines or settling the dust.

All of the above needs to be taken into account during any kind of IR survey. The consensus is that due to all of those difficulties, underground mines need high-quality cameras with a large pixel size and large sensor temperature range. However, even then, their effective range is usually much shorter than outside, and the quality of data decays quickly [1,5].

*2.3. Study Area*

Lubelski Węgiel Bogdanka S.A. Mine is located in the Central Coal Region (CRW) located in the northeastern, best-explored part of the Lublin Coal Basin. Geographically, the Central Coal Region lies within Polesie Lubelskie region, and only small fragments of it extend into the Lublin Upland region. In terms of administration, the Bogdanka mine is located in the Lubelskie Voivodeship, in the area of the Puchaczów commune. Hard coal excavated and sold by LW Bogdanka is used primarily for the production of electricity and heat. Bogdanka's clients are mainly industrial companies, primarily entities operating in the power industry, located in eastern and northeastern Poland. However, since the start of the War in Ukraine, the importance of this mine has grown [20,46].

The mine currently excavates in three Mining Areas: Puchaczów V" with an area of approximately 73.36 km$^2$, "Stręczyn" with an area of approximately 9.38 km$^2$, "Ludwin" with an area of approximately 78.67 km$^2$. It is also conducting roadway works in order to reach and open a new "Cyców" Mining Area, with an area of approximately 40.73 km$^2$. The mining areas of the mine are divided into three mining fields—the Bogdanka field, the Stefanów field, and the Nadrybie field. In each of the fields, there is a pair of shafts: ventilation shaft and downhill and material shaft.

The primary type of excavation is longwall. This is how the mine excavation was planned from the start, and this seems to be the most suable method. However, due to the fact that in case of Bogdanka, coal seams are usually thin (in height), in extreme cases being around 90 cm of usable coal, an introduction of a plow, a version of a longwall shearer suitable for excavation in low seems, was necessary. This was an important step for Polish mining since, for almost 30 years, plows were not used in Polish mining. This was causing a loss of many usable deposits, and the introduction of this technology in Bogdanka led other mining companies to implement this method. There is currently 2–3 excavation frons running simultaneously at the Bogdanka coal mine.

The study area in the mine was located in a low, longwall excavation that was a part of the coal deposit, measuring 1.30–1.40 m in height. This excavation was a so-called 'closing wall' where all the other excavation fields had already been excavated and had collapsed. This type of longwall needs to be run fast, or it will collapse due to the lack of support on either side. The survey was performed in December during the national festive day dedicated to the miners called "Babrórka" celebrations (Saint Barbara is catholic patronage of the miners, and 4th of December is considered a miners festival, many mines stop their operation at or around this date). On this day, no excavations were performed.

A fragment of the wall (excavation front) located in KWK Bogdanka was selected as the test object. The long wall is the section of the mine where the direct extraction of the output takes place. Unlike galleries and other access excavations, the longwall does not have sidewalls secured with permanent linings or any other type of support apart from hydraulic, movable supports that hold the roof. The coal seam and other geological layers are clearly visible and accessible. Coal occurs in the seam deposit alternately with coal-bearing shales. Coal seams often have intergrowths (inclusions) of shales. The shale surrounding the coal deposit is considered waste rock. The unavoidable excavation of shale deteriorates the economic indicators of coal mines. There are also carbonized shales with high coal content that can be an additional source of fine coal during coal processing (in so-called preparation plants). For the economics of coal mining, its energy parameters, thickness, and the sum of intergrowth thicknesses are important. In addition, safety standards and the possibility of people working near low walls (e.g., plow walls with a thickness of up to 1 m) should also be taken into account. The thickness of the seam in the launched longwall is monitored on an ongoing basis. Depending on the type of mine, the speed of the longwall advance, and the geological model, the measurement can take place from once every 2 days to 4 times a day. This allows for better planning of further extraction and obtaining information on the amount of useful minerals in the mine's output.

### 2.4. Experiment Setup

The diagram in Figure 2 shows the workflow used during this study. During the experiment, we performed an onside survey of parts of the longwall excavation front. The survey consisted of a few steps;

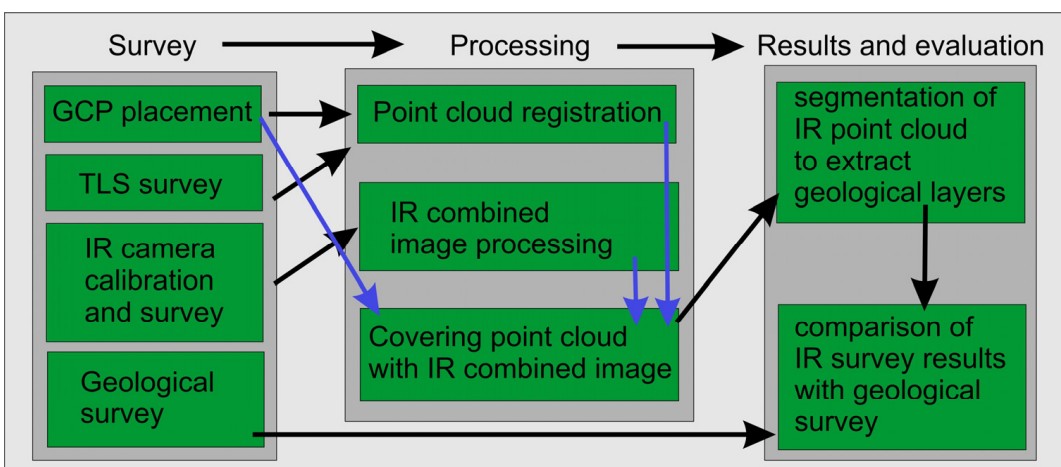

**Figure 2.** Workflow diagram.

1. Placing GCP (ground control points) visible with both necked eye and on IR images for registration (connecting) all obtained data.
2. Geologists performed a standard for this mine geological survey that consisted of cleaning parts of the wall from coal dust (so that they could see the geological layers), measuring their placement in relation to the floor (by putting a 0 of tape measure on the floor and stretching it up as high as they could reach), repeating the procedure in intervals, creation of an interpolated geological layer 2D model.
3. Three-dimensional survey of a longwall, performed with the use of TLS in order to capture the geometry of the longwall.
4. IR survey, images taken with IR camera in order to capture the thermal response of geological structures on the longwall.

The experiment took place on the night of 3–4 December 2019, during the "Barbórka" celebrations when mining operations were suspended, and the number of underground crew was reduced. Only the survey crew was present in the excavation front, and the

machines were not in operation, while the ventilation system remained uninterrupted. As a result of the mining suspension, there was minimal airborne coal dust or pollution. The average geothermal temperature of the Bogdanka mine is 44 °C, and the air temperature during the survey was 22 °C. The study area was a section of the "closing wall" inseam 382, covering an area of approximately 200 m, including 100 m of access gallery and 100 m of longwall.

The FARO Focus 60 laser scanner was used to measure individual point clouds. The point clouds were registered in the local coordinate system using sphere targets and black-and-white chessboards with defined dimensions. Infrared measurements were made using the FLIR S60 camera on two randomly selected fragments of the wall, one of the four meters wide and the other two meters wide.

Seven fragments of shale were spread over an area of four meters by the author, and it was checked beforehand whether they would be visible against the carbon background due to their significantly different emissivity from the surrounding rocks. They were used as a kind of thermal markers (thermal GCP-ground control points) to fit the thermogram into the point cloud [47]. They were also of a different geometry to the rest of the front, so they were easy to identify on the point cloud. Additionally, during the preparation works, ten additional natural GCPs were identified. These were significant changes in geometry or emissivity visible on both thermograms and point clouds. They were usually rocks that fell from the roof or various incursions in the geology. All of those, both natural and artificial GCPs, had to be clearly visible on IR images, point clouds, and visual spectrum images. They were used later in various configurations to create thermal textures

A series of thermograms and photos in the visible spectrum were taken. The accompanying geologist, during the measurement, identified the presence of coal, coal shale, and shale on the longwall. The location of these layers corresponded to the temperature changes on the thermogram. In addition, it turned out that thin intergrowths of shales in the coal were clearly visible. Figures 3 and 4 present a description made by a geologist on a 2-m-long test section of the longwall. Layers of shale, coal, and coal shale are clearly visible on the thermogram but not visible on the point cloud. The geological profile made by the geologist was later used to analyze the quality of the experiment's results. In total, during the 9-h survey, 11 point clouds, 74 visible light images, and 89 IR images were obtained.

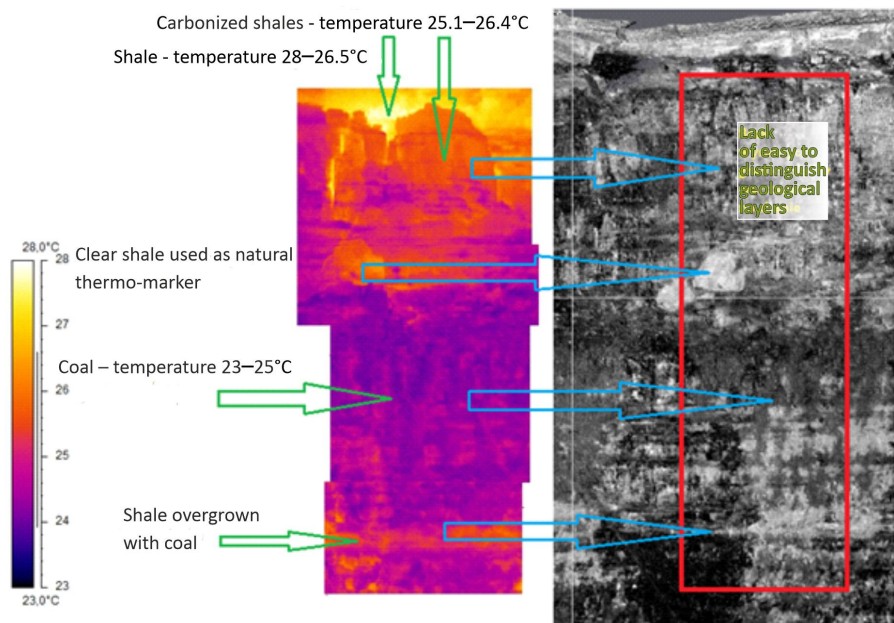

**Figure 3.** On the left—a combined IR image of the part of the excavation front. On the right—a part of the point cloud (in intensity scale) of the corresponding part of the front.

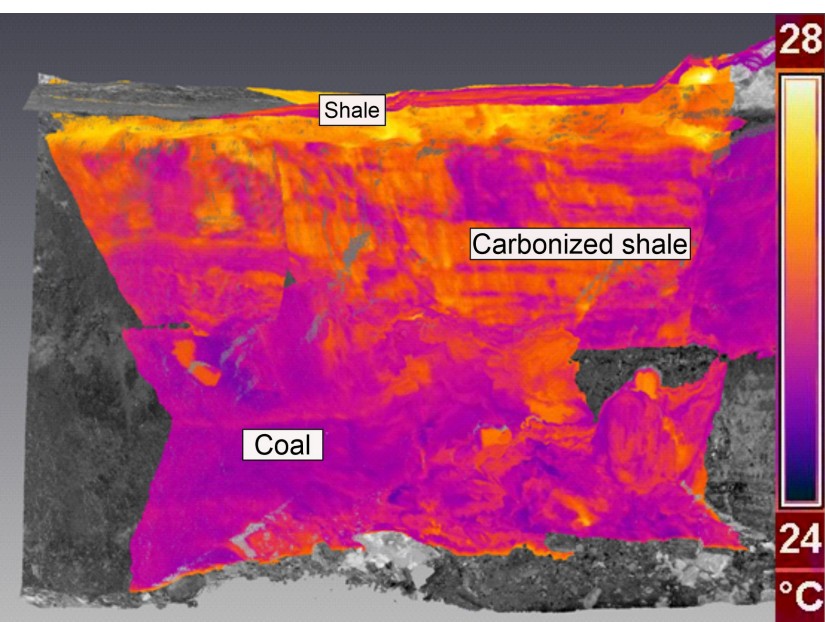

**Figure 4.** A point cloud covered with an IR texture.

### 3. Data Processing

The author conducted an experiment to confirm the usefulness of the Faro SCENE software (https://www.faro.com/en/Industries/More-Industries/Automotive-for-AEC, accessed on 1 January 2023). for covering scans of topographic objects with thermograms [48]. The texturing was performed manually. This allowed us to identify the challenges that IR images present for point cloud-related graphics software and understand the data better, thus proposing a verification procedure. Additionally, it allowed for analyzing and proposing the adjustments that will have to be made in the future to automate the process.

The data processing was carried out in the FARO Scene program. The point cloud used in this experiment was produced by a terrestrial laser scanner (TLS), which uses a laser beam to measure the distance from the scanner to the nearest object that the laser beam reflects from the surrounding [23,49–51]. The FARO Focus is a phase-type scanner that measures the phase of the laser beam on the way back to obtain the distance. The bearing is not measured, but since the scanner moves in the preprogrammed sequence, the bearing of each measured distance is available. With this information, a 3D coordinate of each point that the laser beam reflected from is available, thus creating a point cloud in the local coordinate system with 0,0,0 in the scanning center of the instrument. Usually, a point cloud consists of XYZ spatial coordinates, I intensity (the energy of the laser beam coming back to the scanner), and optionally RGB, the color of each measured point if a camera is available. In order to connect scans made from a few scan stations into one point cloud, a procedure called registration is necessary. Registration can be performed automatically or manually with the use of predefined targets [47]. In our case, we used sphere targets and black and white chessboards of defined dimensions. In total, 11 point clouds were registered, and the average registration accuracy was 13 mm. Each pair of scans was connected with at least four targets, and on average, five targets were used. This is considered adequate in these conditions. From the combined point cloud, the wall itself was isolated, removing fragments of the housing, wiring pipes, and other elements that might obscure the excavation front.

The camera used was a ThermaCAM™ P60 (temperature range −40 to +2000 °C in various settings; thermal sensitivity 0.8 °C; accuracy ±2 °C ±2% of reading, focal plane array (FPA) detector, uncooled microbolometer; 320 × 240 pixels. This is a relatively small pixel-size camera with a high-quality sensor, so it can be used to detect even the smallest

changes in temperature [24]. The first step of processing involved creating a combined image of the excavation front. The decision was made to separate the front into smaller areas and take separate images. The object was photographed in lines with 40% overlap in both directions. IR cameras have high distortion parameters, so a large overlap was useful for further processing. The camera model provided by the producer was later used to remove the distortions, which fixed the images but introduced some quality issues. The images were then cropped on the sides, removing 25% on each side, and matched with each other to create an orthomosaic. An orthomosaic was created with a significantly higher pixel size and limited distortions than one image could deliver (Figure 3). The combined image was compared to the IR image of the entire part of the longwall to see if any easy-to-notice errors were visible. Then, the combined image was saved with some metadata that Faro Scene software would require to process the image further.

The Faro Scene program requires the user to indicate at least six pairs of points corresponding to each other on the image, in this case, the thermogram and the point cloud. According to the instructions, increasing the number of these points and distributing them evenly over the entire scan area should increase the accuracy of the fitting. However, the irregular shape of the object and the relatively small number of pixels on the combined thermal image significantly hinder the operation of the program. The algorithm used, most likely, has not been tested on irregular objects. If more than seven pairs of points were chosen, the match does not work correctly. There was an irregular stretching of the IR images observed. The work came down to locating as many points as possible and adjusting their placement, then reducing their number to seven and trying different combinations to provide a regular placement of adjustment points. The process was very time-consuming. Nevertheless, the effect of the work (Figures 4 and 5) shows the geological structure mapped in 3D the thermal spectrum. The 3D impression facilitates the reception of data by potential users. What is more, in such a model, it is possible to measure the thickness of individual layers.

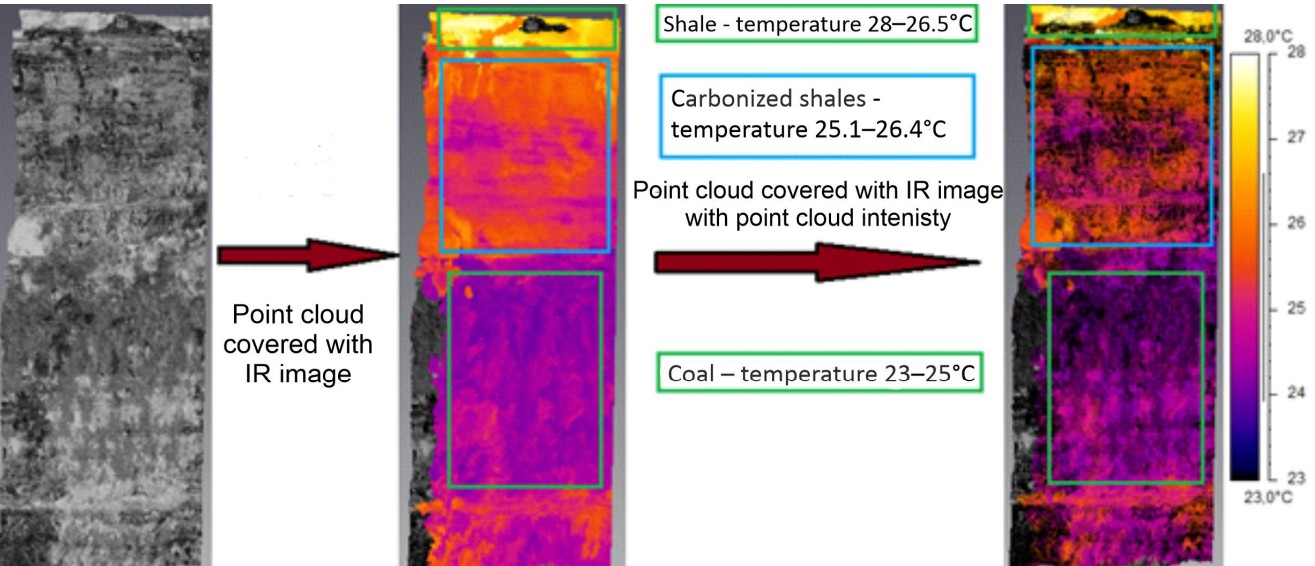

**Figure 5.** Visualization of the results of the survey. A point cloud covered with IR texture with a visualization of I-intensity. Allowing for better 3D view of the longwall fronts structure.

The author decided to conduct a descriptive analysis of the accuracy of the fitting problem. The thickness of layers is examined by a geologist using a classical ruler. The observer takes the measurement himself, holding the ruler against the wall with one hand. The accuracy of such a reading is estimated to be 4–6 cm. In addition, the placement of the 0 of the tape is hard to analyze because it is 'as deep as the tape will go to the floor'. The point cloud measurement and registration combined error is 1.3 cm, so it can be neglected.

Fitting the thermogram into the cloud depends on the accuracy of selecting the adjustment points and the fitting algorithm. Due to the large differences in temperature of the elements and their image on the cloud, the author estimates that this error should not be more than 2–3 cm. The transformation error, image to the point cloud, given by the software was at 1.9 cm.

The projection of the image itself is the least studied part of the analysis. However, when comparing the thermogram with the thermal image superimposed on the point cloud, no fitting errors were noticed. The author estimates the error of such projection to be a few centimeters, in accordance with fitting transformation (image to point cloud) results given by the software. This accuracy is sufficient when profiling the wall. The measurement carried out in this way ensures the safety of the observer, limiting their approach to the unprotected fragment of the roof.

### 4. Results

As a result of this study, a 3D thermal model of two fragments of a long wall was obtained. This model was significantly more detailed and accurate than the sketch provided by the on-site geologist. As a result, a separation of the layers (Figure 6) was produced and presented to the mine, allowing for more detailed calculations of the product excavated from this long wall. Since the process is fully digital, it can be easily integrated into the pipeline used by the mine for scheduling excavation and blending.

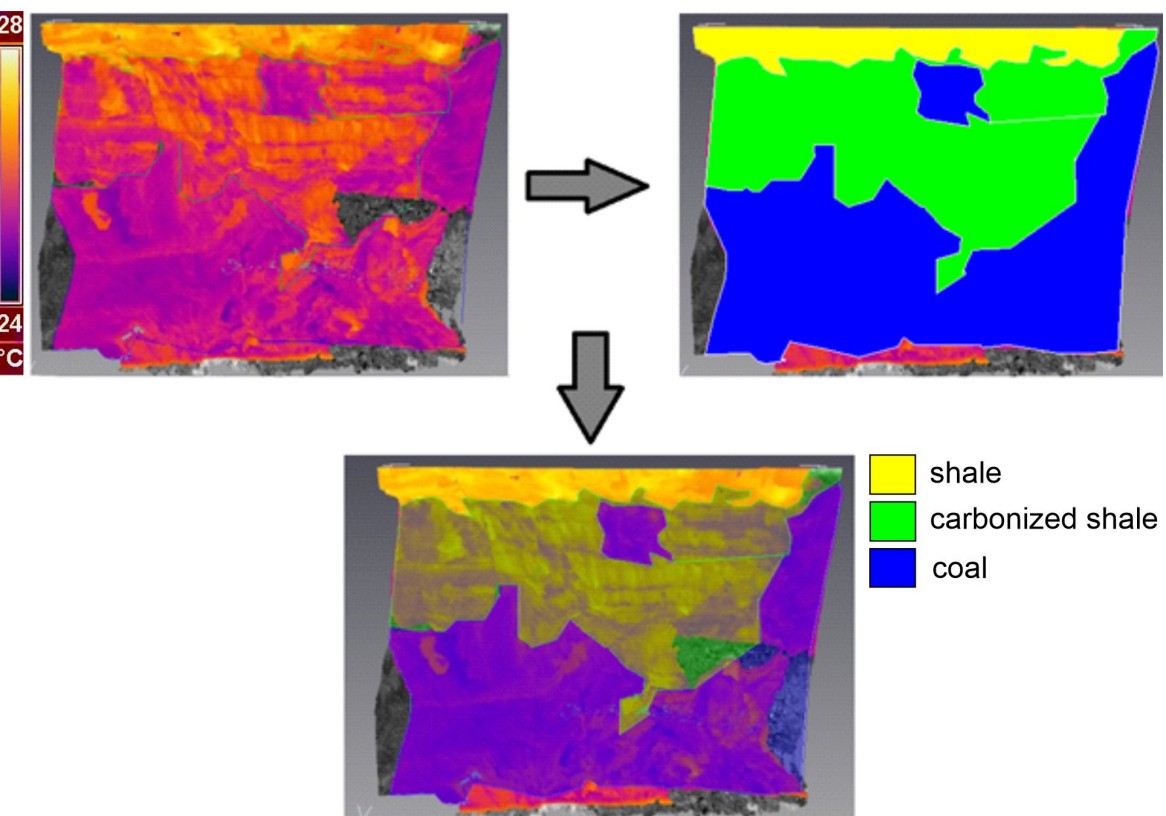

**Figure 6.** Separation of the material at the front of excavation.

The underground mining industry in Poland has been undergoing a fast change in recent years, where it has moved from analog and manhandled management of the mine to the more efficient digital database and scheduling software management. As such, it was necessary to move as much previously analog data to the digital form and also to create new ways of documenting that would allow for an instant introduction of data to the digital database. For much of the data, it was relatively straightforward since sensors

were already storing data in digital form. For example, survey data needs to go through the calculation and vectorization process before it can be introduced. However, since this is all performed in digital space, there is still little falsework. However, geological data, obtained in a described in this paper way, is still often processed in an analog format, both data inquisition and data interpolation. This is why the results of the above-described study are a promising step in the automation of data acquisition and processing. What is more, since it is relatively easy to store it in one coordinate system, automatic interpolation and build-up of a full 3D model of the coal deposit during excavation can be achieved.

The results of this study show the structure of the carbonized shale more clearly than under visible light conditions. Figures 5 and 6 display the combination of IR and intensity data, which reveals the amount of actual coal present in the layer and whether it is worth excavating, as well as how much shale can be expected in the mix. This information is valuable since it provides more data for the preparation plant, which will be blending the material to even out the quality of the final excavation product.

## 5. Discussion

The IR survey takes a few minutes and is not affected by the mine's environment, allowing for remote geological inventory without crossing the safe, roof-supported area. There is also no need for artificial markers to create a 3D textured point cloud. The changing light settings along the long wall do not appear to influence the IR results significantly. However, TLS laser scanning is not optimal in these circumstances, taking too much time and requiring excavation work to stop. Additionally, the manual procedure for creating a 3D texture, although easy once the procedure is in place, takes too much time to perform. This could be mitigated by using a TLS unit with an IR camera, such as the Z + F 5100c with an IR camera mount, but this operation would still take a lot of time and require a relatively long stop of excavation. Moreover, based on the author's experience, the survey's time makes integrating scans performed in changeable conditions difficult, making it imperfect for accurate change detection [17,24].

The limitations of the method will be changeable environmental conditions in the mine. Although at the excavation front, a constant air draft should be maintained, this is not always the case. Additionally, the amount of coal dust may vary over time. All of this can create slight differences in evidence on some parts of the long wall and, thus, degrade the results. Another issue is the procedure of cooling down the face and the machines, as well as the removal of coal dust. This is performed by spraying water onto the face and the machines. Since this process is not consistently carried out, it needs to be taken into account during further inspections using IR cameras.

## 6. Conclusions and Future Works

For future works, the use of mobile scanning systems with integrated IR cameras is being investigated. SLAM (simultaneous localization and mapping) technology allows a device to be positioned and navigated through a space based on its surroundings. When combined with LIDAR, a rigged point cloud can be created on the fly that can be post-processed later to a registered point cloud. A scanner combined with an IR camera would involve placing a camera close to the scanner's arm and calibrating the distance between the devices in order to calculate the projection parameters for the image to be used to color the point cloud. A device such as this could be placed on the longwall shearer, and data could be provided simultaneously, or it could be a separate device running on the conveyor. Devices with such sensors already exist, for example, the ELIOS 3 UAV (unmanned aerial vehicle), which is equipped with both a laser scanner and an IR camera (https://www.flyability.com/elios-3, accessed on 1 January 2023. However, the current limitation is the quality of the IR camera. Future work by the author of this study will concentrate on choosing an IR camera that would be small enough and accurate enough to create a 3D textured point cloud and the creation of an independent platform for 3D-IR surveys.

In conclusion, the current geological survey system needs to be changed as soon as possible since it generates danger to the geologist and does not provide continuous, accurate data. Moreover, the delay in data processing and analysis might be costly to the mine because the thinning of the deposit might not be relisted in time. All this makes creating an IR-based survey system necessary in the near future.

**Funding:** This research received no external funding.

**Data Availability Statement:** Not applicable.

**Acknowledgments:** I would like to thank Sławomir Kubiak for helping with the survey and data processing and Artur Dyczko for organizing access to the mine.

**Conflicts of Interest:** The author declares no conflict of interest.

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
