# Peer review of "Use of 3D Inferred Imagining for Detection of Changes in Geology in Longwall-Type Excavation Front"

_remotesensing, doi:10.3390/rs15112884_

Round 1
Reviewer 1 Report
The topic of the study is of some interest to this Journal and to the
scientific community. However, in my opinion, the paper cannot be
published in its current form as it needs a revision in organization and
linguistically too.
1) The introduction should be more streamlined and coherent. The author
should avoid sub-paragraphs.
2) A paragraph "materials and methods" should be created and the current
paragraphs 1.3 and 1.4 as well as 3 and 4 inserted;
3) The results should be discussed in more detail and should be
compared, if appropriate, with similar case studies or applications;
4) In line 240 there is a repetition ("No excavations were made on this
day").
- English absolutely to be improved: more scientific and less colloquial in some passages (example: line 240 "Santa Barbara in the Catholic patronage of miners");
Author Response
York, 14.05.2023
Dear Reviewers
I would like to thank you for all of the work you put into reviewing this paper. I believe this has improved my paper significantly. In the revised version of this paper has been produced in accordance to your suggestions. This file contains answers for the questions included in the reviews. Within the manuscript all added text in given in purple.
Best regards
Paulina Lewińska
Review 1
Q1. The topic of the study is of some interest to this Journal and to the scientific community. However, in my opinion, the paper cannot be published in its current form as it needs a revision in organization and linguistically too.
R1. Thank you for this comment; I applied reorganization in the current version of the paper.
Q2. The introduction should be more streamlined and coherent. The author should avoid sub paragraphs.
R2. Thank you for this comment, the sub-paragraphs have been removed
Q3. A paragraph "materials and methods" should be created and the current paragraphs 1.3 and 1.4 as well as 3 and 4 inserted;
R3. Thank you for this comment ‘Materials and methods’ section has been added. It includes sections 1.3, 1.4, and 3. Section 4 has been renamed Methodology (in accordance to the 2nd reviewers suggestion). I do strongly believe that Experiment description should exist as a separate chapter, since in case of this study, the experiment setup is important.
Q4 The results should be discussed in more detail and should be compared, if appropriate, with similar case studies or applications;
R4. Thank you for this comment. There have been a few papers published that use IR sensors that use IR sensors for guiding longwall shearers. This is a similar application, however since guiding requires real time feedback and also works on the basis of detection and not modelling. Thus this is not suitable for direct comparison.
However, since we had a geologist present during the experiment, the direct comparison of the geological results with our results have been performed. Geological data have been used for both initial understanding of IR data and for analysis of accuracy of obtained data. This information can be found in L75-95, L267-271 and L380-390 of the current manuscript.
Q5. In line 240 there is a repetition ("No excavations were made on this day").
R5. Thank you for this comment. This has been removed and the sentence rewritten, since I have noticed that some edits have not been applied.
Q6. English absolutely to be improved: more scientific and less colloquial in some passages (example: line 240 "Santa Barbara in the Catholic patronage of miners")
R6. Thank you for this comment, the paper undergo an extensive language check.
Reviewer 2 Report
This study aims to determine whether it is possible to use a thermal imaging camera in the conditions of the mine environment in the longwall excavation area and whether it is possible to distinguish shale interbedding from a coal deposit on the thermographs obtained in this way.
This study has good application value and can reduce the safety risks of geologists.
However, this manuscript has many areas for improvement, and it is particularly necessary to describe the specific details of this method.
Specific Points:
1. Line 27: "infrared (IR) images" should be" infrared images (IR)".
2. Line 31:need to add the full name of the abbreviation UAV.
3. Section 3. "Experiment setup and execution" suggest changing to "Methodology".
4. Section 1.3. Suggest moving to Section 3.
5. Section 2. Suggest supplementing the research area distribution map.
6. Section 4. The description of data processing methods is too general, and it is recommended to supplement the flowchart and schematic diagram.
7. The key issue to be addressed in this article is to distinguish shale interbedding from a coal deposit on the thermographs. However, no comparative experiments, no scientific evaluation of experimental results.
8. Figure 5. No scale, legend, or other elements were found.
9. Section 6. There is no specific analysis of imaging factors in this article's method. No applicability analysis of the method presented in this manuscript.
Minor editing of English language required.
Author Response
Dear Reviewers
I would like to thank you for all of the work you put into reviewing this paper. I believe this has improved my paper significantly. In the revised version of this paper has been produced in accordance to your suggestions. This file contains answers for the questions included in the reviews. Within the manuscript all added text in given in purple.
Best regards
Paulina Lewińska
Review 2
Q1. This study aims to determine whether it is possible to use a thermal imaging camera in the conditions of the mine environment in the longwall excavation area and whether it is possible to distinguish shale interbedding from a coal deposit on the thermographs obtained in this way.This study has good application value and can reduce the safety risks of geologists.
R1. Thank you for this comment.
Q2. However, this manuscript has many areas for improvement, and it is particularly necessary to describe the specific details of this method.
R2. Thank you for this comment, adjustments to the manuscript have been done. In particular, lines L262-278 have been added in order to explain the survey procedure and further processing.
Q3. Specific Points:
- Line 27:"infrared (IR) images" should be" infrared images (IR)".
Thank you for this comment, this has been changed
- Line 31:need to add the full name of the abbreviation UAV.
Thank you for this comment, this has been added
- Section 3. "Experiment setup and execution" suggest changing to "Methodology".
Thank you for this comment, this has been changed.
- Section 1.3. Suggest moving to Section 3
Thank you for this comment, in accordance to suggestions from the 2nd reviewer current version of the
- Section 2. Suggest supplementing the research area distribution map.
Thank you for this comment, however I am not certain if this would add any new information, since this would only mirror the information from existing images.
- Section 4. The description of data processing methods is too general, and it is recommended to supplement the flowchart and schematic diagram.
Thank you for this comment, a flowchart has been added.
- The key issue to be addressed in this article is to distinguish shale interbedding from a coal deposit on the thermographs. However, no comparative experiments, no scientific evaluation of experimental results.
Thank you for this comment. There have been a few papers published that use IR sensors that use IR sensors for guiding longwall shearers. This is a similar application, however since guiding requires real time feedback and also works on the basis of detection and not modelling. Thus this is not suitable for direct comparison. However, since we had a geologist present during the experiment, the direct comparison of the geological results with our results have been performed. Geological data have been used for both initial understanding of IR data and for analysis of accuracy of obtained data. This information can be found in L75-95, L267-271 and L380-390 of the current manuscript.
- Figure 5. No scale, legend, or other elements were found
Thank you for this comment, the description of the elements on the figure, have been moved to the figure .
- Section 6. There is no specific analysis of imaging factors in this article's method. No applicability analysis of the method presented in this manuscript.
Thank you for your comment. I believe that description of applicability can be found in future works section that directly follows discussion. Applicability is described when combining the proposed method with mobile mapping SLAM scanners. At this stage the author is in the process of devising a platform that could be installed on the longwallsheres that would provide data for creation on thermal point cloud. The current limitations are the choice between real time data and post processing. Also storage and introduction of data onto existing infrastructure is a real issue. However I do believe this is more suitable to be described fully in a separate paper.
Unfortunately I do not understand what ‘imaging factors’ mean in this context, this is why I am unable to provide an answer to this comment.
Round 2
Reviewer 2 Report
1. Suggest merging sections 2 and 3.
2. The discussion is too weak, and further optimization is recommended.
3. Line 219-221,“km2” should be “km2”.
Minor editing of English language required.
Author Response
York, 29.05.2023
Dear Reviewers and Editor
I would like to thank you for all of the work you put into reviewing this paper. I believe this has improved my paper significantly. In the revised version of this paper has been produced in accordance to your suggestions. This file contains answers for the questions included in the reviews. Within the manuscript all added text in given in purple.
Best regards
Paulina Lewińska
Review 2
Q1. Suggest merging sections 2 and 3.
A1. Thank you for this comment, the sections have been merged.
Q2. The discussion is too weak, and further optimization is recommended.
A2. Thank you for this comment, and additional paragraph has been added.
Q3. Line 219-221,“km2” should be “km2”.
A3. Thank you for this comment, this has been changed.